# A Review of Agent-Based Programming for Multi-Agent Systems

**Rafael C. Cardoso *** and **Angelo Ferrando ***

Department of Computer Science, The University of Manchester, Manchester M13 9PL, UK
* Correspondence: rafael.cardoso@manchester.ac.uk (R.C.C.); angelo.ferrando@manchester.ac.uk (A.F.)

**Abstract:** Intelligent and autonomous agents is a subarea of symbolic artificial intelligence where these agents decide, either reactively or proactively, upon a course of action by reasoning about the information that is available about the world (including the environment, the agent itself, and other agents). It encompasses a multitude of techniques, such as negotiation protocols, agent simulation, multi-agent argumentation, multi-agent planning, and many others. In this paper, we focus on agent programming and we provide a systematic review of the literature in agent-based programming for multi-agent systems. In particular, we discuss both veteran (still maintained) and novel agent programming languages, their extensions, work on comparing some of these languages, and applications found in the literature that make use of agent programming.

**Keywords:** agent-based programming; multi-agent systems; agent programming languages



## 1. Introduction

Multi-Agent Systems (MASs) [1] are a well established branch of Artificial Intelligence (AI). Even though they are relatively young with respect to more archetypal research areas, MASs have a rich history; in 1995 [2] agent technology was recognised as a rapidly developing research area and one of the fastest growing areas of information technology. Such a statement is still true nowadays, where one can find plenty of research articles, tools, and conferences whose aim is to advance the research in the area. Despite this, MASs are not as widely used as they could be. Considering the agent programming aspect of MASs, according to [3], the key reason is that there is little incentive for developers to switch to current Agent Programming Languages (APLs), as the behaviours that can be easily programmed are sufficiently simple to be implementable in mainstream languages with only a small overhead in coding time. This, amongst the presence of too many unorganised options available, does not help agent-based programming languages and tools to be picked from non-expert users.

An intelligent agent [4] can be generalised as a computerised entity that: is able to reason (rational/cognitive), to make its own decisions independently (autonomous), to collaborate with other agents when necessary (social), to perceive the context in which it operates and react to it appropriately (reactive), and finally, to take action in order to achieve its goals (proactive). An agent-based (or agent-oriented) system is a system where the agents are the main entities, treated as first-class abstractions. From a programming perspective, the same reasoning can be followed. In particular, by using a comparison, we can say that agents are to Agent-Oriented Programming (AOP) languages as objects are to Object-Oriented Programming (OOP) languages. In an agent-based programming language, agents are the building blocks, and programs are obtained by programming their behaviours (how an agent reasons), their goals (what an agent aims to achieve) and their interoperation (how agents collaborate to solve a task).

Agents are well-suited to be used in applications involving distributed or concurrent computation or when communication is required between different components. For this reason, agent technology is useful in applications that reason about messages/objects

received over a network. By preserving their processing state and the state of the world around them, agents are also ideally suited to automation applications. Moreover, autonomous agents can operate without user intervention and can be used in applications such as plant/process automation, workflow management, robotics, and others. Another advantage of agent-based programming is that due to the reasoning cycle present in agents, it is also possible to provide explanations about the decisions that an agent has made.

The aim of this paper is to review the latest work in agent-based programming, to help both experts and non-experts users having a better grasp over the current state of the art in agent-based programming technologies, and to identify future directions of research in this area. In particular, we focus on the latest agent programming languages, platforms and frameworks for the development of MASs. Both theoretical and practical papers are taken into account, and we also briefly discuss recent extensions, existing comparisons, and applications of agent-base programming.

With respect to other reviews and surveys on APLs in literature [3,5–10], our review focuses on recent developments and considers works presenting new APLs, as well as works focusing on extending or comparing existing APLs. Moreover, we consider both theoretical and practical aspects; this helps to have a better understanding of the reality gap between theoretical and practical APLs. Finally, with respect to previous reviews, we focus on the entire class of APLs, and not on a specific area of APLs as in [6], where only the engineering aspects are considered, or in [7], where only APLs platforms are considered, or in [8], where only agent-based simulation literature is analysed, or in [3,5], where their focus is in a specific model of agency (Belief-Desire-Intention—BDI).

This paper is structured as follows. A brief history on agent-based programming is given in Section 2. In Section 3, the systematic review process followed in this paper is presented. Section 4 contains the review findings with the papers found in our search of the literature. In Section 5, the review's results are discussed and future directions are suggested. Finally, in Section 6, the conclusions of the paper are reported.

## 2. History on Agent-Based Programming

In 1993, Agent-Oriented Programming was first introduced [11] as a specialisation of Object-Oriented Programming. Most notably, it discusses the notion of the mental state of an agent, consisting of its information, decisions, and capabilities. This work also describes agent programs in the AGENT-0 interpreter (implemented in the Lisp language) and their communication using speech act theory, the latter is still used to define agent communication in several contemporary agent programming languages. Over the years, many reasoning and cognitive models have been developed for agent-based programming. In this section, we discuss three particular models that have been fundamental in the design of many agent programming languages in the past and that are still being used in new languages these days: Procedural Reasoning System (PRS), BDI, and Situation Calculus.

The Procedural Reasoning System (PRS) [12] (implemented in Lisp) defines a system capable of reasoning about processes, that is, procedural forms of knowledge. An agent in this system is then able to use these procedures to select intentions for achieving particular goals. Unlike in conventional programming languages, these procedures are not invoked a priori, but they are triggered when they are able to contribute towards some goal or to react to some situation. While sharing some similarities to AI planners of the time, its main difference is that it performs partial hierarchical planning in the sense that it interacts with a dynamic environment during the reasoning process, instead of generating a plan for a static environment.

The Belief-Desire-Intention (BDI) model [13,14] consists of a reasoning process that aids the decision-making of selecting an appropriate action towards the achievement of some goal. Its three mental attitudes are: belief—knowledge that the agent believes about its environment, itself, and other agents; desire—the desired states that the agent wants to achieve; and intention—a sequence of steps towards the achievement of a desire. These mental attitudes respectively represent the information, motivational, and deliberative

states of the agent. The workflow in a generic BDI system is shown in Figure 1 and works as such: a belief revision function receives input information from the environment (e.g., sensors), and it is responsible for updating the belief base. This update can generate more options that can become current desires based on the belief base and the intentions base. A filter is responsible for updating the intentions base, taking into account its previous state and the current belief base and desire base. Finally, an intention is chosen to be carried out as an action by the agent. BDI is the most popular model of agency, it has been and continues to be used in many agent programming languages. AgentSpeak(L) [15] is a language that serves as an abstraction of implemented BDI systems that can be used to interpret agent programs as horn-clause logic programs. The theory behind this language has been implemented as a basis for many APLs.

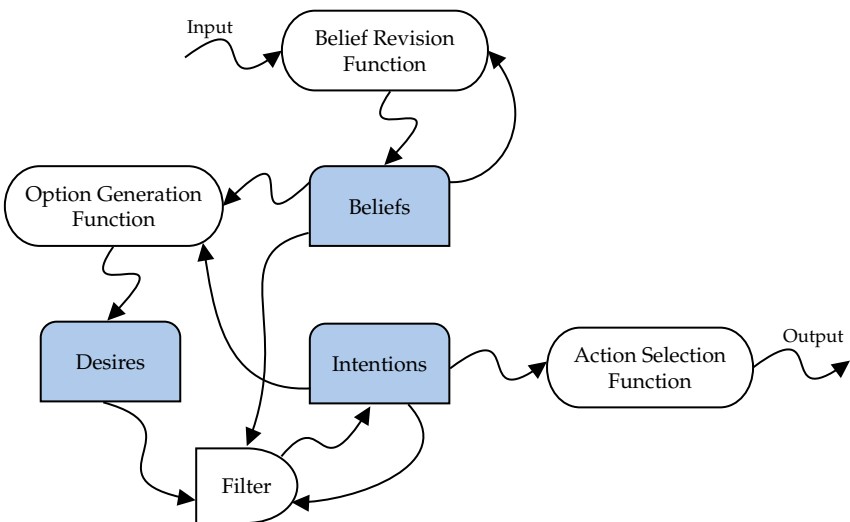

**Figure 1.** The BDI model.

Situation Calculus [16] is a first order language designed to represent changes in dynamic environments. A situation is a first order term representing a sequence of actions. An initial situation is when no actions have occurred yet. The function $do(a, s)$ results in a successor situation to $s$ after executing the action $a$, similar to state transition systems. Dynamic environments play an important role in agent-based programming, and as such, Situation Calculus has been used to model how the world changes as a result of executing actions.

As we will see in Section 4, there are many other models that have inspired agent-based programming languages, however, these three were the most influential in the past history of agent-based programming. Some agent languages share similarities or even mix concepts from other programming paradigms, such as procedural, imperative, object-oriented, functional, actor, concurrent, and so on. Comparing the differences or going into detail about these other paradigms is out of scope of this review, but we still consider agent languages that combine concepts from different paradigms.

Historically, agent-based programming has been used in a myriad of practical applications, such as distributed control of electric power grids [17], governance of room allocation [18] and of automated machine-to-machine applications (e.g., traffic redirection) [19], and detecting privacy violations [20]. In Section 4.4, we cover the more recent attempts in using APLs for programming practical applications.

### 3. Review Methodology

We performed a systematic review of the literature in agent programming languages over the past 5 years (2015–2020). A diagram illustrating our review methodology is shown in Figure 2. For each searched term, we considered the first 10 pages (100 entries) retrieved by Google Scholar (ordered by relevance), for a total of 400 entries (including duplicates).

The terms used in our search were:

- Agent-Based Programming Languages
- Agent-Based Programming Extensions
- Agent-Based Programming Comparison
- Agent-Based Programming Applications

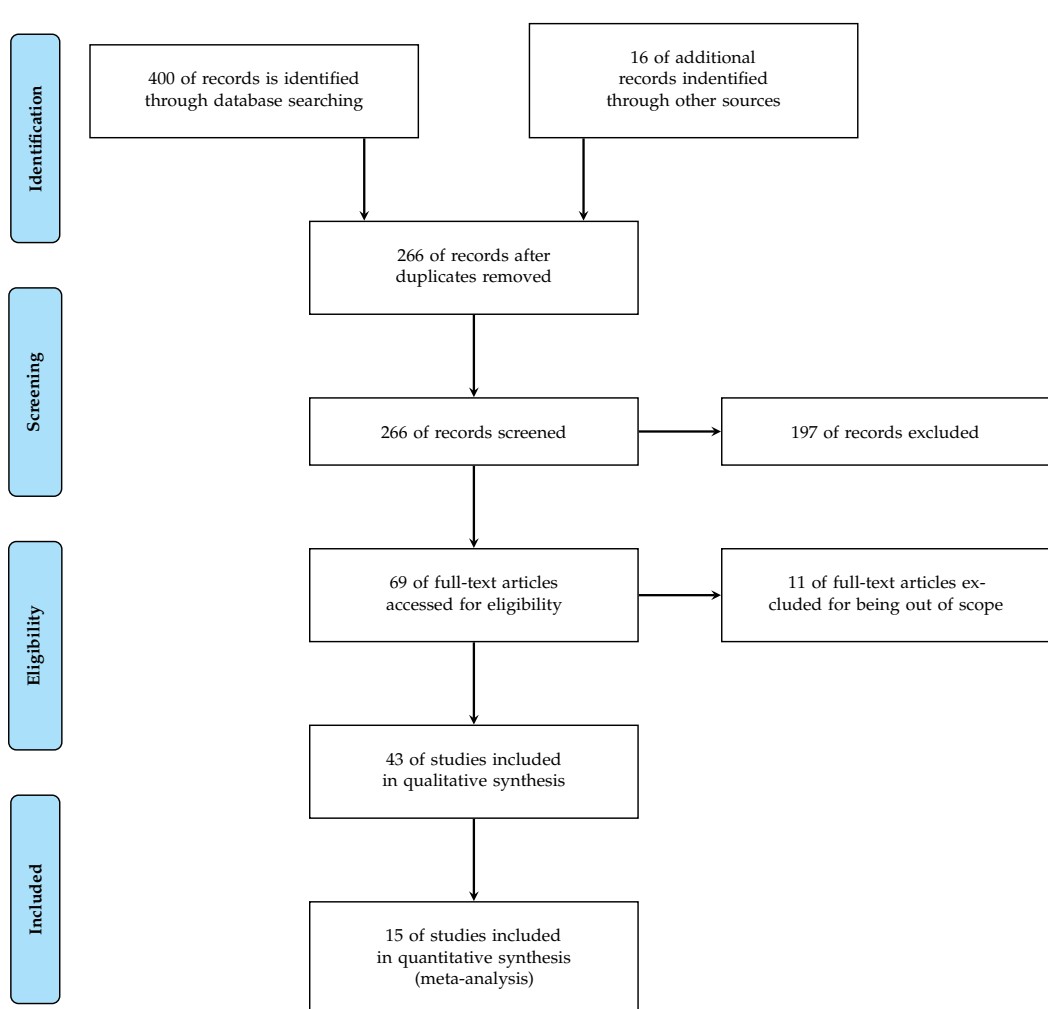

**Figure 2.** Systematic review flow diagram.

After removing duplicates, we had 250 remaining papers. To these, we added 16 entries from external sources; mostly old references that would not appear in the search, plus a few others found in paper citations and other sources. Out of the 16 external entries, eight were influential APLs that have been developed before 2015 and have been updated recently (i.e., 2017–2020).

## 4. Review Findings on Agent-Based Programming for Mas

In this section, we cover all of the research found in our systematic review of the literature. We start with the agent programming languages and their extensions, then continue to discuss the existing comparisons in agent-based programming, and close the section with a brief review of applications in the area.

### 4.1. Agent Programming Languages

We report the agent programming languages found in the systematic review in Table 1. As we alluded to in Section 2, we can see that BDI is clearly the most popular model of

agency, being used in 7 out of the 15 languages. The majority of the implemented languages have been implemented in Java, most likely to take advantage of the cross-platform of Java through its Java Virtual Machine. The table contains only the high-level general-purpose languages that can be used to develop domain independent MASs. While other approaches such as Agent-Based Modelling and Simulation (ABMS) and cognitive agents in robotics are not included in the table, we briefly report some of the novel research that has been done in those areas further below.

**Table 1.** A collection of recent (or recently updated) agent programming languages. Languages that have no publicly available implementation are represented with ✗. In case there are multiple implementation branches, the Last Updated column refers to the last update in the master branch.

| APL | Model | Implementation (Language-Link) | Last Updated |
|---|---|---|---|
| ASTRA | BDI | Java https://gitlab.com/astra-language | 6 November 2020 |
| Chromar | rule-based | Haskell https://github.com/azardilis/Chromar | 14 June 2020 |
| GOAL | rule-based | Java https://goalapl.atlassian.net/wiki/spaces/GOAL/overview | 15 December 2020 |
| Gwendolen | BDI | Java https://github.com/mcapl/mcapl | 7 December 2020 |
| JaCaMo | BDI, organisation, environment | Java https://github.com/jacamo-lang/jacamo | 20 September 2020 |
| JADE | FIPA | Java https://jade.tilab.com/ | 8 June 2017 |
| JADEL | DSL, interaction | Java/Jade ✗ | ✗ |
| Jadex | mixed, BDI and OOP | Java https://github.com/actoron/jadex | 10 January 2021 |
| Jadescript | DSL, scripting | Java/Jade ✗ | ✗ |
| Jason | BDI | Java https://github.com/jason-lang/jason | 12 November 2020 |
| LightJason | BDI | Java https://github.com/LightJason/ | 29 December 2020 |
| PLACE | BDI, HTN | ✗ | ✗ |
| PLASA | Wait-Look-Compute-Move | Java ✗ | ✗ |
| RMAS | database-centric, CPS | Matlab/SQLite ✗ | ✗ |
| SARL | DSL | Java https://github.com/sarl/sarl | 4 January 2021 |

### 4.1.1. General-Purpose APLs

A survey on agent-oriented programming from the software engineering perspective can be found in [6]. One of the main challenges reported in the survey for APL developers is the need to bridge the cognitive gap that exists between the concepts underpinning mainstream languages and those underpinning AOP. In [21] the authors try to fill this gap focusing on understanding the relationship between AgentSpeak(L) [15] and OOP with the goal of trying to reduce the perceived cognitive gap. Such a work proposes a new statically typed agent programming language entitled ASTRA.

In [22] the authors present Chromar, a rule-based notation with stochastic semantics yielding a continuous time Markov chain. Chromar is embedded in Haskell, this gives

it an increased expressive power, and fits with the availability of rich types. In Chromar, rules are first-order abstractions that can both describe a (possibly partial) behaviour of an individual agent and a synchronised action of two or more agents.

GOAL [23] is a declarative agent programming language that uses knowledge base beliefs and goals to support the decision-making of its cognitive agents. Despite sharing similar concepts with the BDI model (beliefs, desires/goals), the GOAL language is more centred towards rule-based decision-making. Agents programs are written in GOAL's specific syntax, but the knowledge of the agent (e.g., rules) are usually represented in Prolog.

Jason [24] is an extension of the AgentSpeak(L) language, based on the BDI agent model. Agents in Jason react to events in the system by executing actions on the environment, according to the plans available in each agent's plan library. One of the extensions in Jason is the addition of Prolog-like rules that can be added and used in the belief base of agents.

The JaCaMo platform [25,26] is composed of three technologies, Jason, CArtAgO [27], and Moise [28], each representing a different abstraction level. Jason is used for programming the agent level, CArtAgO is responsible for the environment level, and Moise for the organisation level. JaCaMo integrates these three technologies by defining a semantic link among concepts in different levels of abstraction (agent, environment, and organisation). The end result is the JaCaMo MAS development platform. It provides high-level first-class support for developing agents, environments, and organisations, allowing the development of more complex multi-agent systems.

Gwendolen [29] initially started as a small subset of Jason in the hopes of developing verifiable agent programs, but has since grown into its own syntax and semantic. Because it is a language that has been built to support agent verification from the ground up, it is limited in what features it can support, however, the basics of AgentSpeak(L) and BDI are all present. There is a vast literature in verification of agent programs and MAS, but we consider them out of scope for this review. Gwendolen, apart from being verifiable, is still a viable language for developing general-purpose MASs.

JADE [30] is an open source platform for the development of peer-to-peer agent based applications. Besides the agent abstraction, it also provides: task execution and composition model, peer-to-peer agent communication based on asynchronous message passing, and a yellow page service that supports the publish and subscribe discovery mechanism. JADE-based systems can be distributed across machines with different operational systems, and has been used by many languages (e.g., Jason and JaCaMo) as a distribution infrastructure.

In [31] the authors present JADEL (JADE Language), an extension of JADE that provides support for the construction of agents and MAS on top of JADE without having to use Java directly; subsequently, in [32], the authors present Jadescript, an extension of JADEL. Jadescript is characterised by a strong expressive syntax largely inspired by modern scripting languages in order to promote readability and to make agent programs more similar to pseudocode.

Jadex [33] allows the programming of intelligent software agents in XML and Java. The agent abstraction is based on the BDI model, and provides several features such as: a runtime infrastructure for agents, multiple interaction styles, simulation support, automatic overlay network formation, and an extensive runtime tool suite.

LightJason, a highly scalable Java-based platform for BDI agent-oriented programming and simulation is presented in [34]. LightJason is based on a logic language which extends AgentSpeak(L) with lambda-expressions, multi-plan and -rule definition, explicit repair actions, multi-variable assignments, parallel execution, and thread-safe variables. Even though the language is inspired by AgentSpeak(L) and Jason, LightJason is implemented from scratch.

In [35] an AOP language called Planning based Language for Agents and Computational Environments (PLACE) adds AI planning capability to agents. PLACE has a syntactic structure close to BDI, while the planning is done in a Hierarchical Task Network

(HTN) planner. In contrast to other AOP languages, actions in PLACE have durations associated to them, thus, requiring the planner to be able to handle temporal information. Agents in PLACE have the ability to recover from failures by adapting their activities to the new situations. For this purpose, a plan repairing mechanism is added that repairs a plan if the unanticipated events in the environment cause the plan to become unfeasible.

In [36], the Programming Language for Synchronous Agents (PLASA) is proposed. PLASA is platform-independent and facilitates a rapid implementation of co-operative applications on multiple physical robots and in dynamic environments. Essentially, PLASA implements a variant of the Wait-Look-Compute-Move model proposed in [37], where robots move synchronously. It is designed as a high-level programming language, which allows users to specify the instructions to be performed by robots in a human-readable language.

Relational Model Multi-Agent System (RMAS) [38] is a database-centric approach for multi-agent systems suitable for the embodiment of reasoning and control in Cyber-Physical Systems (CPS). Initial implementation of RMAS is proposed by the coupling of the Matlab environment and the SQLite database language.

SARL's [39] focus is to provide an extensible language that is equipped with the minimum amount of concepts (i.e., key concepts) required to support AOP. The language aims to provide abstractions for concurrency, distribution, interaction, decentralisation, reactivity, autonomy, and dynamic reconfiguration. To do so it is not based in any model, but instead it creates its own Domain-Specific Language (DSL) in order to provide a reduced and more lightweight core.

### 4.1.2. ABMS, Robotics, and Others

As recognised in [40], there is a gap between Agent-Oriented Software Engineering (AOSE) methodologies and the development of ABMS. To overcome this issue, in [41] an AOSE process called Process for Developing Efficient Agent-Based Simulators (PEABS) is proposed. It uses the INGENIAS methodology [42] for modelling the specification and designing its structure. It applies an adaptation framework that allows ABMS developers to obtain simulations with a high efficiency for large amounts of data. Another approach for developing ABMS is presented in [43,44], where the authors propose a new cognitive agent architecture based on the BDI model and integrated into the GAMA modelling language [45]. With respect to previous integration works between BDI and ABMS, in [43] the architecture proposed aims to be flexible and easy to use for non-expert users. Another work which aims to integrate BDI and ABMS is presented in [46], where the authors present a framework that allows BDI cognitive agents to be embedded in an ABMS system. Compared to [43], reference [46] is more general since its objective is to integrate any BDI-based system with ABMS. The only requirement is that the percepts (or environmental observations/events) of interest to each agent and the actions that the agent may execute in the simulation environment can be identified a priori.

ALLEGRO (=ALGOL in PREGO [47,48]) is a programming formalism based on belief architecture for stochastic domains which is intended as an alternative to GOLOG [49] for high-level control in robotic applications. Another language which is based in GOLOG and the situation calculus is introduced in [50], where a prototype implementation of Yet Another GOLOG Interpreter (YAGI), an action-based robot and agent programming language, is presented. YAGI offers bindings for popular robotics frameworks such as Robot Operating System (ROS) [51] and Fawkes.

In [52] the authors present a Cognitive Affective Agent Programming Framework (CAAF), a framework based on the belief-desire theory of emotions that enables the computation of emotions for cognitive agents (i.e., making them cognitive affective agents). The authors present semantics showing the programming constructs of these agents. With these constructs, a programmer can build an agent program with cognitive agents that automatically compute emotions during runs.

### 4.2. Agent Programming Languages Extensions

In the previous section, we reported works presenting novel APLs (2015–2020), along with works presenting the most influential APLs ($\leq$2015) that are still maintained. Now, we consider the most influential works that have extended existing APLs. We refer to APL extensions as works that have changed existing APLs internally, either by adding new features or by building on top of existing APLs, for example, by customising the APL for a new and specific scenario.

An enhanced version of Multi-Agent System for Competitive Electricity Markets (MASCEM) [53] is presented in [54]. This extended version of the MASCEM simulator aims at supporting the integration of new and complementary models. The facilitation in accommodating different tools and mechanisms is provided by important structural implementation decisions, making MASCEM able to deal with the constantly changing and highly demanding environment of electricity markets. In particular, the new extension of MASCEM brings the use of ontologies to support players' communications.

In [55], the authors present TABSAOND, an extension of PEABS [41]. The main difference between TABSAOND and PEABS is that the former focuses on the design and implementation of the decision-making processes in non deterministic scenarios. In addition, simulators are now deployed as mobile apps and online tools.

In [56] a conservative synchronisation model is proposed for the SARL language and its runtime platform Janus. Since Janus does not make any assumption on the ordering of the events that are exchanged by the agents, it is not possible to use the Janus platform for agent-based simulation involving time without providing the platform with a specific synchronisation mechanism. A model for such a mechanism is described in their extension.

The authors of [57] propose new programming constructs for integrating an advanced yet rule based emotion model, EMIA [58], in line with the 2APL [59] agent language. The combination of both has been carried out by redefining the syntax, semantics and deliberation cycle of 2APL. This combination mainly focuses on event-based emotion generation, and the resulting simulation shows high believability in the emotions expressed by the agent when responding to the real life scenarios.

ARGO [60] is a customised Jason architecture for programming embedded robotic agents using the Javino middleware and perception filters.

In [61], the authors show how procedural reflection in the agent programming language meta-APL [62] can be used to allow a straightforward implementation of some of the steps in the deliberation cycle of a BDI agent, by allowing both agent programs and the agent's deliberation strategy to be encoded in the same programming language.

An extension [63] to Jason and Gwendolen allows the agents in these languages to communicate with ROS, thus supporting the programming of autonomous agents that can control and perform high-level decision-making in robotic applications developed in ROS. The extension is done through an interface that is used as the environment between the agent and ROS, and the communication between the environment and ROS nodes is performed using the *rosbridge* library. The main difference between their work and past attempts at extending traditional APLs to support ROS is that their approach requires no additional modifications in either of the two APLs or ROS, making it usable and portable to different versions of these tools. Similarly, in [64] a framework for using Jason with ROS in embedded systems is presented and a new architecture is introduced to support lower-level interactions between ROS and the agent.

Finally, in [65] a model for a BDI agent programming framework integrating reinforcement learning and an implementation based on the Jason programming language are introduced. The approach supports the design of BDI agents where some plans can be explicitly programmed and others instead can be learned by the agent during the development/engineering stage.

### 4.3. Agent Programming Languages Comparison

From the research described in the previous two sections we can observe that there are many APLs for developing MAS available in the agent-based programming community. Unfortunately, very often the evaluation of a language is partially or even completely missing. Some studies such as [66] have been done in the past to compare agent languages with other paradigms, in that case the comparison was with actor-based languages. In their results the authors have shown that agent languages (specifically Jason in that work) can indeed be competitive with more lightweight languages such as actors.

In [67], the authors present an evaluation framework for assessing existing or newly defined domain-specific modelling languages for MASs. The evaluation targets both the language and the corresponding tools and provides both qualitative and quantitative results.

A comparison between the pseudocode of a well-known algorithm for solving distributed constraint satisfaction problems and the implementation of such an algorithm in JADEL is shown in [68].

The work in [69] focuses on comparing parallel platforms that support multi-agent simulations and their execution on high performance resources as parallel clusters.

The authors of [70] perform a systematic evaluation of ABMS approaches differentiating the concepts of how complex the model behaviour is and how complicated the model structure is, and illustrate the non-linear relationship between them. Then, they evaluate the trade-offs between simple (often theoretical) models and complicated (often empirically-grounded) models.

### 4.4. Agent-Based Applications

In this section we list some of the latest applications using agent-based programming. Our goal here is to show the wide variety of application domains that agents can be useful in, thus, this list is not exhaustive and not the main focus of our review.

The Multi-Agent Programming Contest (https://multiagentcontest.org/) (MAPC) is an annual international competition that occurs since 2005. Its purpose is to stimulate research in multi-agent programming by introducing complex benchmark scenarios that require coordinated action and can be used to test and compare multi-agent programming languages, platforms, and tools. Implementations using different agent-based platforms and languages have been used in the last few years; such as JaCaMo [71–73], Jason [74,75], GOAL [76].

Agent-based models to simulate and evaluate the transmission of the coronavirus disease (COVID-19) have been proposed in [77,78]. There is an entire research area focused on using agent-based technologies in the energy industry. For example, in [79], MAS technologies are used for the control of Microgrid, its optimisation and market distribution. For further reading, there is a survey [80] on the applications of MAS in the control and operation of Microgrids, and a review [81] of the state of the art in the application of MAS to energy optimisation problems.

In [82], an application of ABMS is presented to study the relationships between human activities and land-use/land-cover changes to support scientific decisions regarding reasonable land planning and land use. The model is implemented based on the Repast modelling platform [83].

In [84], the authors present and illustrate FlowLogo, an interactive modelling environment for developing coupled agent-based groundwater models. FlowLogo is implemented in NetLogo and is the first integrated software offering a straightforward way to represent agent behaviours that evolve with groundwater conditions. A systematic survey on ABMS tools and applications can be found in [8].

A methodological guide to the use of BDI agents in social simulations and an overview of existing methodologies and tools for using them is provided in [10].

Agents can be used for developing self-managed Internet of Things (IoT) systems due to their distributed nature, context-awareness and self-adaptation (for further reading

about using microservices as agents in IoT [85,86]). In [87] the authors aim to enhance the development of IoT applications using agents and software product lines in self-management systems.

In [88], experiments to validate the programming of autonomous robots using Jade-script are presented. It presents the novel support for perception handlers that has been recently introduced in the language to cope with the high data rate of sensors in robotic applications.

## 5. Discussion and Future Directions

As we have shown, there exists a wide variety of options for agent-based programming, from more traditional approaches (e.g., BDI) to simulation or planning-based. Some languages have also attempted to combine concepts from agent-based programming with other programming paradigms, most prominently from object-oriented programming. One of the main drawbacks of trying to achieve a wider community of programmers in agent-based programming is the lack of knowledge and familiarity with its concepts, that are significantly different from other more common paradigms. Agent-based programming languages that use some of the concepts from these other paradigms can help bring new programmers that would otherwise be too intimidated. These hybrid languages have their own niche of applications depending on the concepts that they use, which may sometimes overlap with the more "pure" agent programming languages, but "pure" approaches are still necessary to fully tackle all agent programming abstractions (agent, environment, organisation, interaction, etc.).

Out of the 15 agent programming languages listed in Table 1, five do not have publicly available implementation (i.e., the code is not hosted in a public/accessible domain). This represents a significant issue, as it limits the practical usability of the proposed language and makes it difficult to quantitatively compare other languages against it. Not all extensions to existing languages require an implementation to be useful, however, having one available is always positive for the community.

Even though there are several qualitative (e.g., concepts, features) comparisons in the literature, the use of different models of agency makes it difficult to provide a fair comparison between the features present in these languages. A more in-depth study has to be conducted to identify the fundamental features of agent oriented programming, and more importantly, how these features fit in the different models of agency that existing programming languages use.

Quantitative (e.g., performance) comparisons of programming languages are trickier due to the development cycle of having constant updates, which is even more common in programming languages developed in academia (as most of the agent programming languages are). Nevertheless, it is important to develop agent-specific benchmarks that can be used easily by the community to evaluate new programming languages or extensions to existing languages.

Most languages offer a range of different examples that showcase their features and strengths. While these examples are certainly useful to better understand and learn the language, they are usually not enough to convince new users of the applicability of the language in real-world applications. Complex and realistic case studies are hard to develop, but the agent community has available a suite of complex scenarios as part of the annual MAPC that could be better exploited to test and compare agent programming languages.

Two recent surveys [3,5] focused on BDI agent programming outline the limitations and challenges in the area. In a manifesto [3], the author argues that it is necessary to extend the feature set of current APLs to enable wider adoption of agent technology. The author also disagrees with past surveys that the lack of more polished methodologies and tools is not the main factor (although it does contribute to) in the limited adoption of APLs; instead, the author suggests that there is little to no incentive for developers to make the change to AOP, as the behaviours currently shown in applications from the literature can be implemented in more mainstream languages with limited effort. The survey in [5] recaps

the history so far and the state of the art in agent programming with a focus on BDI-based approaches. They identify as a major challenge for future research the integration of AI techniques in agent programming languages as an important and necessary step to the widespread acceptance and adoption of AOP.

*Recommendation for Further Research*

Considering the past 5 years of research on APLs, many novel frameworks, platforms and models have been proposed. Each one of these, along with new extensions, enriched the agent-based literature and enlarged the spectrum of possible applications. Nonetheless, as rightfully observed in [3,5], the major issue in current APLs is not in their set of features, but in their usability. Generally, there is no desire in learning new languages when the advantages are not straightforward. In our review, we analysed APLs that were both expressive and powerful, but with major usability issues; such as the absence of a (maintained) tool, documentation, and qualitative and quantitative comparisons with other languages. In our opinion, further research on APLs will have to tackle these usability issues in order to try to spread the use of APLs outside the agent community.

## 6. Conclusions

Agent-based programming is a thriving research area of artificial intelligence. In this review paper, we have classified both veteran and recent contributions according to four different categories: agent-based programming languages, their extensions, the comparisons between languages, and finally, some of the applications using these languages. For each contribution we briefly reviewed the content and outlined the key results. To have a better understanding of the current state of art, we did not only focus on the latest approaches, but we also briefly reviewed the most prominent agent-based programming languages that are still being maintained.

Every year there are many extensions to existing languages and even entire new languages being proposed, however, most of them are limited to formal descriptions without any implementation to support the formal theory. The small subset of approaches with implementation lack any effective evaluation. Comparing new approaches to the state of the art is one of the major steps required to advance the area of agent-based programming. Qualitative and quantitative comparisons can help to identify gaps in existing languages, which can lead to either improvements or new approaches that are able to cope with the challenges raised. Moreover, in our review we have also identified a lack of real-world applications. In order to widen the use of these languages, it is important that their usability in the real-world is well documented, thus, we encourage and recommend more application-based papers that can demonstrate features of agent-based programming in the real-world.

**Funding:** This research was funded by the UK Industrial Strategy Challenge Fund (ISCF) delivered by UK Research and Innovation (UKRI) and managed by Engineering and Physical Sciences Research Council (EPSRC) under the Robotics and AI for Extreme Environments programme with grants Robotics and AI in Nuclear (RAIN) Hub (EP/R026084/1), Future AI and Robotics for Space (FAIR-SPACE) Hub (EP/R026092/1), and Offshore Robotics for Certification of Assets (ORCA) Hub (EP/R026173/1).

**Conflicts of Interest:** The authors declare no conflict of interest. The funders had no role in the design of the study; in the collection, analyses, or interpretation of data; in the writing of the manuscript, or in the decision to publish the results.

## Abbreviations

The following abbreviations are used in this manuscript:

| | |
|---|---|
| AI | Artificial Intelligence |
| AOP | Agent-Oriented Programming |
| AOSE | Agent-Oriented Software Engineering |

| | |
|---|---|
| APL | Agent Programming Language |
| BDI | Belief-Desire-Intention |
| CPS | Cyber-Physical Systems |
| DSL | Domain Specific Language |
| HTN | Hierarchical Task Network |
| IoT | Internet of Things |
| MAPC | Multi-Agent Programming Contest |
| MAS | Multi-Agent System |
| OOP | Object-Oriented Programming |
| PRS | Procedural Reasoning System |

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
