# Peer review of "A Review of Agent-Based Programming for Multi-Agent Systems"

_computers, doi:10.3390/computers10020016_

Round 1

Reviewer 1 Report

Intelligent and autonomous agents is a subarea of symbolic artificial intelligence where these agents decide, either reactively or proactively, upon a course of action by reasoning about the information that is available about the world (including the environment, the agent itself, and other agents). The topic of this review paper is timely and interesting. This reviewer only has some minor comments:

The section of Agent-Based Programming Extensions should be improved. The connotation of extension is a bit vague.

The existing results on agent-based programming for multi-agent systems have been reviewed. It’s better to further add some sentences to clearly provide the practical applications for these existing computer-based programming languages.

In literature review, the authors considered the first 10 pages (100 entries) retrieved by Google Scholar (ordered by relevance), for a total of 400 entries (including 115 duplicates). Other than such a review manner, some deep analysis works can be done to analyze the most influential results and then some classification jobs can be performed.

Author Response

We thank the reviewer for their comments and suggestions.

- The section of Agent-Based Programming Extensions should be improved. The connotation of extension is a bit vague.

We added a paragraph to the introduction of Section 4.2 to better explain what kind of extensions we have considered in our review.

- The existing results on agent-based programming for multi-agent systems have been reviewed. It’s better to further add some sentences to clearly provide the practical applications for these existing computer-based programming languages.

Section 4.4 contains the practical applications of APLs which have been published in the past 5 years. We added a paragraph to Section 2 for a general overview of the types of practical applications that agent-based programming has been used for in the past.

- In literature review, the authors considered the first 10 pages (100 entries) retrieved by Google Scholar (ordered by relevance), for a total of 400 entries (including 115 duplicates). Other than such a review manner, some deep analysis works can be done to analyze the most influential results and then some classification jobs can be performed.

We added a sentence to Section 3 to clarify that besides the novel approaches (obtained from the search), we also include the most influential APLs from the past. Determining how influential the new approaches are is not possible at this moment, as APLs require time for being adopted by the community and evolving into more robust techniques. Nevertheless, we provide some basic classification in Table 1.

Reviewer 2 Report

The current paper summarises and reviews agent-based programming investigations for multi-agent systems. The paper considers both veterans (still maintained) and novel agent programming languages, their developments, comparing some of the applied languages, and applications observed in the literature of agent programming. The paper is well-written and well-organised as a review paper. However, it would be great to apply some modifications as follows:

  1. Originality: it is the most important parameter for publishing a top review paper. Please discuss in this importance.
  2. Advances knowledge and original thinking: How to list the challenges and gaps of the previously published review papers and what are the main contributes of this work to provide a better understanding of this certain topic.
  3. It can be useful if we had a subsection of the theory behind the Agent-based Programming model. (adding figures are strongly recommended )
  4. classifying and grouping the exiting frameworks based on some metrics such as popularity, novelty and application help to develop this paper substantially. 
  5. Please add a subsection for your recommendations for further research.

Author Response

We thank the reviewer for their comments and suggestions.

- Originality: it is the most important parameter for publishing a top review paper. Please discuss in this importance.

- Advances knowledge and original thinking: How to list the challenges and gaps of the previously published review papers and what are the main contributes of this work to provide a better understanding of this certain topic.

We added a paragraph in the Introduction section to better explain in which way this review is different from similar ones in the literature (and as such, what is the originality of our review paper).

- It can be useful if we had a subsection of the theory behind the Agent-based Programming model. (adding figures are strongly recommended )

We added a figure of the standard BDI model in Section 2. As we have shown, BDI is the most popular model of agency. We hope this figure will help conceptualise how agent-based programming works. We feel that adding formal theory would not be very helpful here since languages use different models and have their own operational semantics.

- classifying and grouping the existing frameworks based on some metrics such as popularity, novelty and application help to develop this paper substantially.

Our classification is based on the four main terms that we reported in the review. For each term, we reported all the relevant works published in the past 5 years on that area. Notions such as popularity, and novelty are hard to quantify; nonetheless, in some sense, every work we cited that has not been published in the past 5 years can be considered "popular" (the veteran APLs), while the rest of the works can be considered "novel" (the recently proposed APLs). We added a sentence to Section 3 to clarify this. Applications in the last 5 years are reported in Section 4.4, as well as connected to APLs in Table 1. We have also added a paragraph about past applications in Section 2.

- Please add a subsection for your recommendations for further research.

We added subsection 5.1 to summarise our recommendations considering the results outlined in our discussion.